# Experiences of Non-Pharmaceutical Primary Care Interventions for Common Mental Health Disorders in Socioeconomically Disadvantaged Groups: A Systematic Review of Qualitative Studies

**DOI:** 10.3390/ijerph20075237

**Published:** 2023-03-23

**Authors:** Kate Bernard, Josephine M. Wildman, Louise M. Tanner, Akvile Stoniute, Madeleine Still, Rhiannon Green, Claire Eastaugh, Sarah Sowden, Katie H. Thomson

**Affiliations:** 1Population Health Sciences Institute, Newcastle University, Newcastle-upon-Tyne NE2 4AA, UK; 2National Institute for Health and Care Research (NIHR) Applied Research Collaboration (ARC) for the North-East and North Cumbria (NENC), Newcastle NE4 5TG, UK; 3National Institute for Health and Care Research (NIHR) Innovation Observatory, Newcastle University, Newcastle-upon-Tyne NE2 4AA, UK

**Keywords:** systematic review, qualitative research, mental health inequalities, primary care, socioeconomic disadvantage, social prescribing

## Abstract

Common mental health disorders (CMDs) disproportionately affect people experiencing socioeconomic disadvantage. Non-pharmaceutical interventions, such as ‘social prescribing’ and new models of care and clinical practice, are becoming increasingly prevalent in primary care. However, little is known about how these interventions work and their impact on socioeconomic inequalities in health. Focusing on people experiencing socioeconomic disadvantage, this systematic review aims to: (1) explore the mechanisms by which non-pharmaceutical primary care interventions impact CMD-related health outcomes and inequalities; (2) identify the barriers to, and facilitators of, their implementation in primary care. This study is a systematic review of qualitative studies. Six bibliographic databases were searched (Medline, ASSIA, CINAHL, Embase, PsycInfo and Scopus) and additional grey literature sources were screened. The included studies were thematically analysed. Twenty-two studies were included, and three themes were identified: (1) agency; (2) social connections; (3) socioeconomic environment. The interventions were experienced as being positive for mental health when people felt a sense of agency and social connection. The barriers to effectiveness and engagement included socioeconomic deprivation and underfunding of community sector organisations. If non-pharmaceutical primary care interventions for CMDs are to avoid widening health inequalities, key socioeconomic barriers to their accessibility and implementation must be addressed.

## 1. Introduction

The treatment of common mental health disorders (CMDs) such as depression and anxiety form a significant part of the primary care workload. In the United Kingdom (UK), 40% of general practice (GP) appointments have a mental health component [1], and in many high-income countries, most people with a mental health problem are seen only in primary care [2,3,4].

The association between socioeconomic disadvantage and CMDs is well established [5]. Low income, debt, unemployment, and low social class are all associated with higher rates of mental health problems, and those experiencing socioeconomic disadvantage suffer disproportionately from CMD-related adverse health outcomes [6,7,8,9]. In addition to experiencing higher rates of CMDs, there is evidence that people living with socioeconomic disadvantage are less able to access and benefit from the treatments for these difficulties [10].

Medications such as antidepressants are sometimes effective in the management of CMDs. However, there are growing concerns about the over-prescription or inappropriate use of pharmaceutical interventions [11,12]. Antidepressant prescribing has increased over the past three decades—a trend that has outpaced the rise in CMD prevalence [13,14]. Given the associations between CMDs and socioeconomic disadvantage, there are concerns that the overreliance on pharmaceutical treatments may be ineffective and result in the medicalisation of the distress that represents a normal reaction to adverse social and economic conditions [15]. In the UK, certain indicators of socioeconomic disadvantage, such as unemployment, poor housing, or living in an area of deprivation, are associated with higher antidepressant prescription rates (often independent of a diagnosis of depression) [10,13,16], but poorer access to primary care consultations and other treatments, such as talking therapies [10,17,18].

Non-pharmaceutical interventions represent alternative treatment options for mental distress. Psychological ‘talking’ therapies have become widely embedded in primary care; for example, England’s National Health Service Improving Access to Psychological Treatment (IAPT) service [19]. The IAPT service has been thoroughly evaluated and found to lead to improvements in depression and anxiety; however, there is also evidence that less advantaged patients struggle to access IAPT services [20].

A range of new non-pharmaceutical options for treating CMDs in primary care are now also being introduced. In the UK, ‘social prescribing’, which aims to address the social determinants of mental ill health, is being formally embedded within primary care [21]. Social prescribing enables people to access a range of support, such as housing and financial advice, bereavement support, and arts activities [22]. Primary care professionals can refer directly to community services, but most often social prescribing involves link workers, whose role is to connect patients with sources of support in the community [22,23]. New methods of clinical practice have also been introduced in some areas; for example, integrating clinical psychologists within general practice teams [24]. In addition, ‘new models of care’ for mental health were introduced in NHS England’s Five Year Forward review [25]; these aim to improve care for CMDs by, for example, integrating primary and secondary services.

Non-pharmaceutical interventions are increasingly prevalent in primary care practice. Health inequalities and mental ill health both currently represent key policy objectives in the UK and other high-income countries, particularly post-COVID-19 [26,27]. For these reasons, it is important to review and synthesise the qualitative studies to identify the potential mechanisms by which primary care interventions improve the mental health of people living with socioeconomic disadvantage, as well as the facilitators and barriers affecting people’s engagement with these interventions. A focus on the experiences of people living with socioeconomic disadvantage is necessary to facilitate the design of effective services, to help improve access for marginalised groups, and to mitigate the risk that the interventions may increase health inequalities. This systematic review was undertaken alongside a quantitative review that aimed to synthesise the evidence for the effects of non-pharmaceutical primary care interventions on CMDs in socioeconomically disadvantaged groups [28]. The quantitative data was extracted on to a range of CMD-related health outcomes, including anxiety and depression, distress, wellbeing, self-reported mental health, and healthcare utilisation for CMDs. Together, the review findings will guide the commission of more equitable mental health services.

Focusing on people experiencing socioeconomic disadvantage, this qualitative systematic review aims to: (1) explore the mechanisms by which non-pharmaceutical primary care interventions impact CMD-related health outcomes and inequalities; (2) identify barriers and facilitators to the implementation of non-pharmaceutical CMD interventions in primary care.

## 2. Methods

The full methodology has been previously described in the published protocol [29]. The review was registered with PROSPERO (CRD42021281166) [30]. The methods are reported in accordance with the ‘ENTREQ’ statement for enhancing transparency in reporting the synthesis of qualitative research [31].

### 2.1. Search Strategy and Data Sources

The following databases were searched from inception until 1 June 2021: Medline; ASSIA; CINAHL; Embase; PsycInfo; and Scopus (example search strategy in Appendix A). In addition, the resource list of the Social Prescribing Network [32], the Social Interventions Research and Evaluations Network [33], and third sector websites were purposefully searched for relevant articles. Backwards and forwards citation chaining of the included studies was undertaken and the relevant systematic reviews were assessed.

### 2.2. Screening and Selection

Following de-deduplication, the titles and abstracts were screened, and the potentially relevant full texts were assessed for eligibility (see Table 1). One reviewer screened each record, and a second reviewer checked a random 10% sample at both stages of the screening process. Screening conflicts were resolved via discussion and adjudication by a third reviewer.

Socioeconomic disadvantage was assessed according to individual-level socioeconomic characteristics (e.g., income, employment status) or aggregate area-level measures of deprivation (e.g., Index of Multiple Deprivation score).

Any interventions meeting our stated inclusion criteria were included and we did not exclude any interventions based on an assessment by the authors of the evidence base behind any given intervention.

### 2.3. Data Extraction and Quality Appraisal

The data extraction and quality appraisal were undertaken by one reviewer and checked in full by another. A modified version of the relevant CASP tool was used to quality appraise the included studies [34].

### 2.4. Thematic Synthesis

Thomas and Harden’s [35] three-stage approach to thematic synthesis was used in the data analysis. Using the NVIVO software [36], the text from the findings sections of each study was coded line by line and then grouped into descriptive themes according to the similarities and differences. The analytical themes were initially generated by KB through a reflection and return to the original data. They were adapted iteratively on discussion with the wider team until consensus was achieved.

## 3. Results

### 3.1. Findings from Systematic Searches

Following de-duplication, a total of 2313 studies were identified from the six databases. After screening, 22 qualitative studies met the inclusion criteria (including five employing mixed methods). The characteristics of included studies are shown in. A PRISMA diagram detailing the search and selection process is displayed in Figure 1 [37].

Sixteen studies were rated as being high quality using CASP, as well as three medium and three low. Twelve studies were set in England, six in Scotland, one in Ireland, two in Canada, and one in the United States.

None of the studies focused exclusively on people living with CMDs. Instead, they involved wider groups of service users (such as people living with long-term conditions, multimorbidity, ‘complex needs’, or ‘unmet social needs’) that included some people living with CMDs—most commonly anxiety and depression.

Eighteen studies took place in an area of socioeconomic deprivation. Two studies involved the majority of participants being described as low income, one involved participants who were recipients of Medicaid services in the United States of America (US), and one involved a citizens advice service predominantly accessed by people with financial or employment issues.

Nineteen studies involved social prescribing interventions, of which thirteen involved link workers. Two studies involved new methods of clinical practice, both designed to integrate depression care within the management of long-term conditions. One concerned a programme of Behavioural Activation delivered by practice nurses; the other was a pilot intervention of psychological wellbeing practitioners acting as case managers alongside practice nurses. One study involved a new model of care: a large-scale system redesign to deliver integrated care for those with complex needs on Medicaid in the US (see Table 2).

Individuals participating in an intervention will be referred to as ‘people’ or ‘service users’; those working to directly deliver the intervention as ‘providers’, and healthcare professionals as ‘HCPs’.

### 3.2. Findings from Thematic Analysis

Three themes emerged: AgencySocial connectionsSocioeconomic environment

All three themes can be understood as mechanisms by which non-pharmaceutical interventions impact upon CMD-related health outcomes and inequalities. Each encompasses both the facilitators of, and barriers to, people accessing and engaging with the interventions.

#### 3.2.1. Theme 1: Agency

Agency is broadly understood as individuals having power to make choices and to act on them. The interventions helped people feel that they were able to make their own choices, act independently, and affect change, which was experienced as being positive for their mental health.

##### Sense of Control and Choice

People described the benefits from the sense of control and choice they experienced when engaging with the interventions in which they were given the opportunity to choose activities and dictate the nature and pace of their engagement [39,42,44,45,49,52,57]. For example, a service user taking part in a community hub project explained: 

*“You can come in here [community hub—LifeRooms] for say 10 minutes, 20 minutes, half an hour… you are in charge of what you’re doing. I think it’s really really important and just that little bit of control can make you feel on top of the world”*.[45]

*“Being able to take an active role in decision making was contrasted with interactions with HCPs that often felt more prescriptive and less collaborative”*.[39,49,58]

##### Confidence, Motivation and Purpose

Many people felt that their involvement in the interventions had increased their confidence, particularly in social interactions and physical activity [40,42,44,45,48,50,51,58]. Motivation levels also increased during the interventions [44,49,58], while some people reported a new sense of purpose [39,40,44,50,51]. This often occurred against a backdrop of previous feelings of worthlessness: *“so this is kind of giving me a feeling of, you know, you’re not useless”* [39].

##### Understanding of Health and Self-Management

Some people felt that the interventions allowed them to gain a better understanding of their own conditions, such as enabling the identification of patterns and triggers of spells of poor mental health [39,45,48,49,56,58]. This improved understanding was often discussed in parallel with the development of condition-management [39,40,45,47,49,53,56,58]; for example, the recognition of mood patterns, allowing for better communication of needs [45]. Some interventions improved the insights and coping strategies for both mental and physical health conditions [39,45,56,58]. Indeed, a number of people had gained insights into the links between their physical and mental health, both in terms of how physical health impacted upon mental health (e.g., recurrent urinary tract infections contributing to low mood) [49,58], as well as vice versa (e.g., effect of emotional eating on diabetes control) [39].

##### Functioning and Achieving Goals

People felt that their involvement in the interventions helped them to structure their days, develop routine, and achieve daily activities and functions [39,40,45,48,51,56,58]: *“I am out of bed, I am dressed, depending on the day whether I’m able to cope with the shower, but certainly washed, dressed and today I am out the house, I am here”* [45]. Managing everyday tasks could help people regain a sense of normality, which for some was seen as central to ‘recovery’ [48]. Some of the interventions resulted in enjoyment, relaxation, and respite from other life challenges [39,44,48,50,51]. Several people felt able to look towards wider long-term goals—most commonly education and employment [40,42,44,45,48,51,58]. Support from the providers played an important role in this: *“An’ I’m contemplating maybe college next year, but I actually feel safe in the fact o’ doing that because I know I’ve got [the link worker] helping me”* [44].

Improved control, confidence, purpose, motivation, self-management of health, and functioning can all be understood as contributing to individuals gaining or regaining a sense of agency—the power to make choices, and to act on them. However, for some people, factors hindering a sense of agency acted as barriers to accessing and engaging with the interventions. Poor health and a lack of awareness around the interventions curtailed people’s power to make choices and act upon them, thereby acting as barriers to their sense of agency (see below).

##### Health Conditions

Some people who would have otherwise engaged in the interventions felt unable to do so due to their health [42,44,49,58]: *“I want to do stuff but my body won’t let me”* [44]. This included physical health conditions [42,44], as well as the symptoms of CMDs, such as low energy associated with depression:
*“You get the days where you are feeling dead down and you can’t be bothered…There are things I want to do, but over these last few months I just haven’t had the energy.”*.[58]

##### Lack of Awareness

Many people did not know about the interventions or understand why they had been referred, which may have prevented them from making informed choices about their engagement [38,40,42,43]. This was also commented on by GPs: *“they [patients] still, similar to me, to begin with, don’t really understand the idea of social prescribing”* [42].

#### 3.2.2. Theme 2: Social Connections

The interventions helped people to connect with others, to feel cared for, and to feel less alone.

##### Shared Experience and Community

Sharing problems with others who had been through similar experiences helped people to feel less alone [39,44,45,51,58]: “*I think when you meet people… who also have had problems, it kind of reminds you that you’re not alone and that there’s hope because… you’re all in it together*” [51]. For example, a member of a bereavement group gained comfort from seeing others who had also lost their partners but were “further on” and coping well [51]. Meeting others and having a sense of community was also widely experienced as being beneficial to mental health [38,39,42,44,45,48,49,50,58]. People felt that others cared about them, were looking out for them, that they belonged to a group, and could connect. For one participant, the simple experience of sharing a meal with another person was emotionally significant:

*“I met someone there and we clicked and we had lunch in the little coffee shop there and it was like oh my god this is the first time in my adult life I have sat and had lunch with a friend”*.[45]

##### Information Sharing

The benefits of information sharing were discussed by the service users, HCPs, and providers. Information sharing was facilitated by the providers working in teams alongside the HCPs in the general practice setting [41,47,56], as well as administrative aids—such as advice workers having access to medical records in order to help the service users with application forms [54]. The HCPs widely valued the intervention providers’ expertise on social interventions as being beneficial to holistic care [41,43,47,50,54,56]. For the service users, information sharing helped them feel their care was joined-up:

*“I think it’s a good idea that they should know that you’ve got a bit of depression because when I go in there and she says your blood sugar, I said well, I’ve been a bad boy, I’ve eaten this, that and the other, she shouldn’t start saying, oh, what are you doing that for?”*.[47]

##### Positive Relationships

Their relationships with the intervention providers, and the social prescribing link workers in particular, were frequently described as being central to people’s experiences of the intervention. People reported that the providers listened to them, understood them, and cared about them [38,39,42,43,44,45,49,50,51,52,53,57,58]. The perception that the providers had time to spend with them was contrasted with medical consultations, who frequently felt rushed [38,45,57,58]. Accepting and non-judgemental attitudes were particularly valued [39,45,48,50,57,59]: 

*“Another thing that I find for which I’m very grateful and surprised is how understanding people here are. It’s about one of the very few places that I feel welcome and respected as I am”*.[39]

Indeed, a couple of service users viewed their relationship with the providers in terms of companionship that was otherwise lacking for them [58,59]. The providers’ practical support, such as accompanying service users to activities and appointments, was also widely appreciated [39,42,44,46,49,51,52,53,58,59].

The GP/patient relationship also facilitated engagement with the interventions, with several people feeling encouraged to participate in the interventions recommended by a trusted GP [46,52]. Others emphasised the acceptability of the interventions that took place in the setting of a GP surgery. For some, this was due to convenience [43,56], but was more commonly discussed as being due to privacy and the avoidance of the stigma related to mental health difficulties [43,46,47,54]: 

*“It’s very much more acceptable for our patients to see someone regarding mood at the practice as opposed to going externally to see a counsellor… more acceptable when it’s seen to be the nurse… People like to hang hooks on names, patients don’t generally go round talking about their depression, but you do hear them going around all the time talking about their diabetes or angina or whatever”*.[56]

However, such preferences were not universal, and some participants preferred keeping physical health management settings separate from mental health (Knowles et al., 2015).

##### Difficulties in Relationships with Providers

A small number of people described difficulties in their relationships with the intervention providers [38,55,58]. Strong service user/provider relationships, when disrupted, could lead to disengagement, with service users feeling abandoned when the providers moved on [55,58]: 

*“My first worker left, I used to see her a lot. I was put onto another one… Now she’s left and they’ve put me onto somebody else who I’ve never seen…I just feel as though I’ve been let down… pushed to one side”*.[58]

Dependency on link workers, given the intense nature of the support role, was also identified as a difficulty by link workers themselves [59].

#### 3.2.3. Theme 3: Socioeconomic Environment

Some of the interventions enabled improvements in people’s socioeconomic circumstances, which was experienced as being positive for mental health. For others, their socioeconomic circumstances presented a barrier to engaging with the interventions. On an organisational level, the socioeconomic environment could present a barrier to implementation.

##### Help with Social Needs

People found help with finances—including benefits, debts, and returning to work—extremely valuable [43,49,55,58,59]. Some described the palpable impact this had on their mental health:

*“Whatever money I owed like electricity and TV licence was in my mind always eating me from inside. I sorted out that and it just changed so many things … It changed my attitude, it changed my behaviour and it changed my mood … I am not depressed like before”*.[49]

Some of the link workers viewed helping with social needs as a central aspect of their role [59]. GPs also described adapting their practice to better cater to the needs outside of the biomedical model [41,42]:

*“My consultation had to move away from this sort of biomedical model of depression…and see the patient a bit more in the wider family and community… Talk about appetite or sleep leads quite smoothly to antidepressants and sleeping tablets but doesn’t move to social prescribing… So I had to change my consultation style … I find it easier now”*.[42]

##### Socioeconomic Deprivation

Low income could act as a barrier to engagement with interventions. One study reported that a social prescribing intervention was viewed as being similar to schemes designed to remove people from receiving state benefits, and thus was seen as a potential threat to welfare entitlement:

*“Now there is a big drive to get people off benefits—rightly or wrongly—so they are reluctant to go anywhere where it says Job Centre Plus. I think they (patients) saw a little bit of similarity between social prescribing and Discover Opportunities and that put the brakes on”*.[42]

Low incomes made accessing the interventions problematic [43,58]. Poor transport access in rural areas was described as contributing to a “cycle of deprivation” for those who could not travel outside of the area for work, compounding their mental health problems [43]. Others were put off accessing the suggested interventions that were not affordable or did not offer enough free sessions [46,52].

##### Social Needs beyond HCP Capacity

Some HCPs did not view social concerns as being part of their role *“Is it really my job to refer people to the Citizens Advice Bureau? Is it really my job to tell them about their weight? They’re social things …we don’t really have the expertise”* [57]. More often, HCPs felt they did not have the power or capacity to address wider social problems and were already overwhelmed by other demands [40,57]. This included a sense that social problems were too complicated and that it was too late for HCPs to fix them: *“We have a problem with underage drinking … as doctors I’m not sure that there’s hellish much that I can do, by the time people walk in through my door it’s probably too late”* [57]. HCPs also expressed a lack of awareness of social sources of support [41,42,57,58].

##### Feeling Failed by the System

For several people, feeling failed, let down, and excluded by both the healthcare and welfare systems affected their willingness to engage in the interventions [44,47]. People described experiences of being ‘passed’ around between doctors’ appointments and the Job Centre (UK government offices providing advice for the unemployed) without feeling heard or helped by either, causing hesitancy to engage further in the system:

*“There’s nobody for the likes o’ me. You’re just left tae, I don’t know, vegetate… No, the system’s totally wrong… As I say, they told me… “You’ll never work again.” So, where do you get the hope fae? Where dae you get the faith fae?”*.[44]

##### Sustainability and Funding

People felt concerned about the short-term nature of the interventions, and what would happen afterwards [40,44,52,53], including concerns about being “hung out to dry” when the programme ended [53]. For one service user, worries about the programme’s sustainability became a new source of mental distress: *“I’ve got a big fear that [the Links Worker Programme] stops. That is actually part o’ my anxiety*” [44]. Funding problems were also frequently discussed as a barrier to implementation—mostly by the providers and HCPs in relation to community sector programmes [38,42,55,56]. This was closely tied to sustainability, in that the programmes were not guaranteed to last due to constantly “chasing funding” [55]. One GP discussed the risk of unstable programmes (often shutting down quickly due to short-term funding) leading to GPs’ reluctance to refer people only for them to experience further disappointment by the system [42].

Link workers were also described as having stretched capacity in their roles, as well as being limited in their ability to refer onwards to community organisations due to their reduced availability [55,59]. There were resulting concerns about the quality of the services provided due to the demand being unmatched by resources:

*“I’m seeing more and more of the time, the resources demand, the stretch on organisations in terms of the amount of people that seem to be getting referred to these organisations now. And I think potentially the quality of service of these organisations could suffer”*.[55]

The above concerns were frequently discussed in the context of austerity [41,55,58,59]—an economic policy, prominent in the UK from 2010, that cut social spending and increased taxation, resulting in reduced public and community services [60].

#### 3.2.4. Interactions between Themes

Figure 2 illustrates the interactions between the three themes. Agency and social connection are understood as being central mechanisms by which people experienced positive mental health outcomes from the interventions. These two mechanisms can be interpreted as being mutually reinforcing: it is likely that individuals’ sense of agency helped to enable social connection, and vice versa. These mechanisms can be understood as being influenced by the ‘professional environment’ (for example, the roles of HCPs, and the professional relationships between the HCPs and providers), which in turn created barriers and facilitators to positive outcomes. At the wider level, the outcomes were influenced by the socioeconomic environment in which the individuals and the healthcare system operate—with poverty, organisational funding and resources affecting both the professional environment and individual mental health. These interactions suggest that people’s agency (ability to make choices and act independently) was influenced by socioeconomic forces, which shape the choices available to people.

## 4. Discussion

Non-pharmaceutical approaches to treating and managing CMDs are increasingly prevalent in primary care. People living with socioeconomic disadvantage experience the highest burden of CMDs, and it is therefore vital that these interventions are acceptable and accessible to this group of service users. Our review aimed to explore the mechanisms, barriers, and facilitators by which non-pharmaceutical primary care interventions impact on CMD-related health outcomes for people experiencing socioeconomic disadvantage. Twenty-two papers were included in the review, and three themes emerged from the analysis.

The interventions were experienced as being positive for mental health when people felt they could gain a sense of agency and social connection. People living with CMDs in more deprived areas are also more likely to have more complex health needs, including comorbidities [10]. Many of the interventions included in this review had benefits beyond improving mental health, resulting in the improved management of physical comorbidities. The key barriers to positive outcomes included individual socioeconomic disadvantage affecting accessibility and wider socioeconomic factors affecting implementation.

Our findings on the importance of agency and social connections support those of the previous reviews of social prescribing interventions across a range of patient groups [22,61,62,63,64]. For example, Tierney et al. [63] draw on theories of ‘social capital’ and ‘patient activation’ as mechanisms by which link worker programmes had an impact, whilst Pescheny et al. [62] emphasise ‘self concepts and feelings’, ‘day-to-day functioning’, and ‘social interactions’ as key elements of service users’ experiences of social prescribing. Moreover, several of the primary studies reviewed found value in the ‘self-determination theory’ to explain the intervention impacts based on the psychological need for autonomy, competence, and relatedness [39,44]. The importance of emphasizing basic human relationships and social connection in supporting people’s health and social needs also echoes previous work by Cottam [65] on re-imagining the models of healthcare and the welfare state.

In contrast with previous evidence, our review finds explicit evidence of the *wider* socioeconomic environment affecting intervention impact—going beyond the level of the individual. On an individual level, we find that the support to address social needs was important for improving mental health, whilst socioeconomic disadvantage negatively affected engagement. On a wider level, we find that the resource constraints linked to austerity affected the ability of the community sector organisations to implement interventions. Link workers were overstretched, often unable to refer people on due to the demand being unmatched by resources, and funding for programmes was discussed as being short term and unpredictable. It is worth noting that in the context of mental ill health and pre-existing disadvantage, uncertainty around funding is likely to be particularly harmful for this group of service users.

Given the majority of studies reviewed were set in the UK after 2010, the context of austerity represents a crucial backdrop to these findings. The instability of the community sector, as well as people’s sense of being failed by ‘the system’, should arguably be understood through a wider structural lens. This lens takes into account how political and economic choices, and the distribution of power and resources in society, have a significant effect on people’s health. Our review supports the idea that addressing the mental health needs of those at the sharp end of severe social and economic inequality requires a decided emphasis on the structural determinants of health [23].

### 4.1. Strengths and Limitations

To the best of our knowledge, this is the first review to synthesise the literature on the experiences of non-pharmaceutical primary care interventions for CMDs in socioeconomically disadvantaged groups. The findings provide useful insights into the mechanisms by which non-pharmaceutical primary care interventions may improve people’s mental health, as well as the facilitators of, and barriers to, people’s engagement with these types of intervention. A strength of using a qualitative design was that we were able to uncover some of the complexities involved in the delivery and experiences of interventions in this space [66].

Our study has a number of limitations. Due to resource constraints, and a desire for a generalisability of findings across similar primary healthcare settings, only studies conducted in high-income countries were included. We may have missed valuable insights into non-pharmaceutical interventions in low- and middle-income countries. Important findings from studies in other languages may also have been missed due to only including literature published in English.

The current evidence base is also limited. While we set out to explore a range of non-pharmaceutical interventions, most of the included studies focused on social prescribing, making it harder to draw conclusions about new models of care and new methods of clinical practice. Although all of the studies involved one or more participants with CMDs, none of them focused exclusively on CMDs—instead involving people with a wider range of health and social needs. This limits our ability to draw conclusions about CMDs specifically, with the findings likely to instead be more generalised. Similarly, although all of the studies involved a majority of participants living with socioeconomic disadvantage, several also included a minority of participants who were not, which may have diluted our findings with regard to this specific subgroup. All of the studies focused on the experiences of people who had actively engaged with interventions. We are therefore likely to be missing important perspectives on the barriers to access and engagement, which was often mentioned as a limitation by the authors themselves [49,58]. Finally, we originally intended to review several dimensions of inequality, as described in the protocol, but focused on socioeconomic disadvantage only due to resource constraints.

### 4.2. Implications

Our review underlines several gaps in the literature on non-pharmaceutical interventions for CMDs in primary care. Firstly, we recommend a need for more research involving participants who did not engage with interventions, or those who only engaged briefly. This would help to discern more accurately the barriers to access and engagement, and further inform policies to reduce inequalities in this area.

Secondly, we recommend further research exploring other dimensions of disadvantage and marginalisation, beyond a socioeconomic focus. Given that significant inequalities in mental health exist according to gender [67], race and ethnicity [68,69,70], age [5], disability status [71], and sexual orientation [72], a better understanding of the mechanisms, barriers, and facilitators of non-pharmaceutical interventions in a range of marginalised groups is needed.

Thirdly, we recommend further research focusing on new models of care and new methods of clinical practice for CMDs in primary care. An improved understanding of these types of interventions is likely to become increasingly necessary as their prominence increases.

Our review emphasises that interventions that are difficult to access for those most in need of them, which risks widening health inequalities. We also bring to light systemic problems with funding for the community sector organisations—an integral part of delivery for the interventions reviewed—linked to austerity. Finally, the review highlights a general lack of awareness of non-pharmaceutical mental health interventions in primary care. The implications for policy are summarised as follows:If existing inequalities are to be seriously addressed, the experiences of those living with socioeconomic disadvantage must become a central part of decision-making processes.For interventions to be effective, the community sector organisations they rely upon need to be adequately and sustainably funded., This is likely to require pressure on policy makers to provide sustained funding for community organisations that provide non-pharmaceutical primary care interventions for CMDs.Raising awareness of non-pharmaceutical interventions among both healthcare professionals and service users is likely to be beneficial for engagement and implementation.

## 5. Conclusions

As non-pharmaceutical interventions for mental health become more embedded in UK primary care, there is a need to consider the experiences of people living with socioeconomic disadvantage as a central part of policy decision-making, in order to avoid widening the existing health inequalities. Future research should focus on exploring the barriers to accessibility for those who have not engaged with interventions, as well as exploring the experiences of other disadvantaged and marginalised groups. There is also a need for sustainable funding of the community sector organisations that are frequently relied upon to deliver such interventions. Addressing the mental health needs of those affected by social and economic inequality requires a decided focus on the structural determinants of health.

## Figures and Tables

**Figure 1 ijerph-20-05237-f001:**
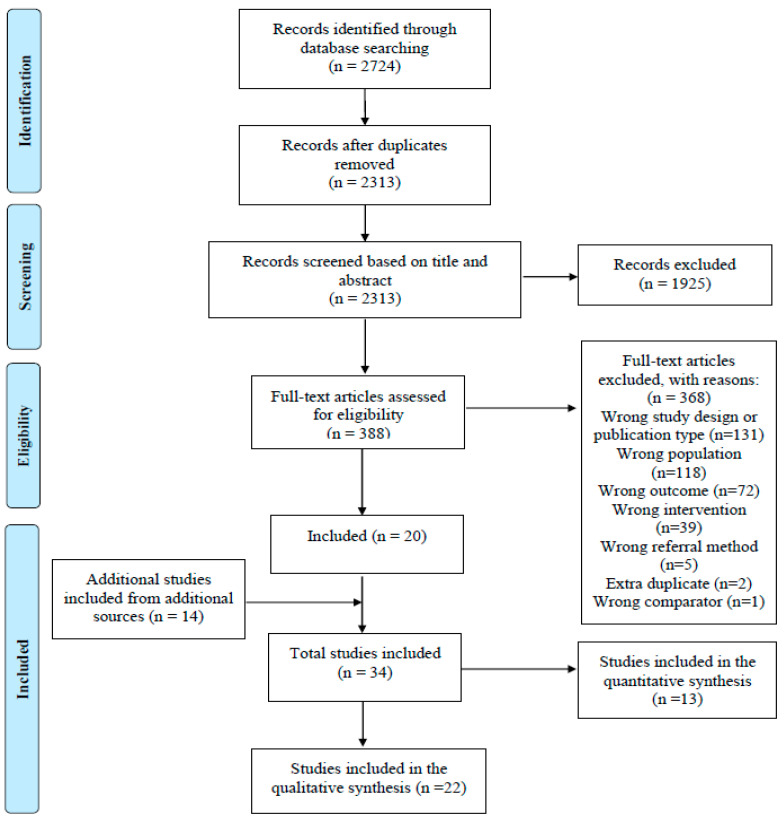
PRISMA diagram [28].

**Figure 2 ijerph-20-05237-f002:**
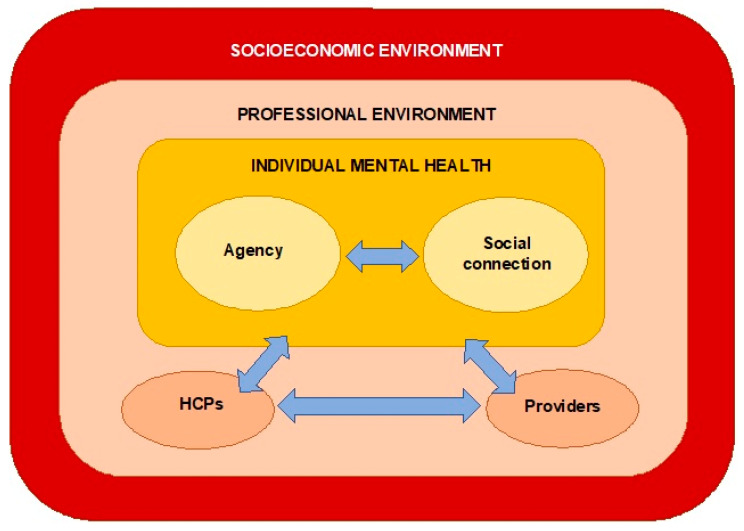
Interactions between themes.

**Table 1 ijerph-20-05237-t001:** Eligibility criteria.

Inclusion Criteria	Exclusion Criteria
All or some participants with one or more of the following: anxiety; depression; somatoform disorders; post-traumatic stress disorder; and post-natal depression. All or the majority of participants from a socioeconomically disadvantaged population group. Adults receiving treatment in primary care in any high-income country. Any interventions involving social prescribing, newmodels of care, or new methods of clinical practice, delivered or referred from primary care teams. Qualitative and mixed method primary studies. Studies published in English language in an OECD high income country.	People with the following conditions: psychosis; dementias; child and adolescent mental disorders; conversion disorders; body dysmorphic disorders; personality disorders; eating disorders; suicide; self-harm; substance use disorders; intellectual disability; epilepsy; and developmental disorders. Studies exclusively investigating the effects of pharmaceutical interventions. Editorials and letters.

**Table 2 ijerph-20-05237-t002:** Characteristics of included studies (* FGD = Focus Group Discussion).

Author, Year and Setting	Aim	Methods	Participants	Health Conditions Targeted	SES Characteristics	Intervention Description	Quality Appraisal
Bertotti et al., 2018 [38] City and Hackney, London, England	To explore what worked in the social prescribing pilot in City and Hackney, for whom and under what circumstances.	Mixed methods: Quantitative online surveys, qualitative interviews, FGDs *, observations of sessions with social prescribing coordinators	Service users (n = 17), community organisations (n = 3), social prescribing coordinators (n = 3), commissioners (n = 2), GPs (n = 2)	Social problems or mild to moderate mental health problems, including anxiety and depression	Area of socioeconomic deprivation	Social prescribing: link workers 23 GP practices; patients referred to social prescribing coordinators who referred to community organisations.	Medium
Bhatti et al., 2021 [39] Ontario, Canada	To explore how social prescribing as a process facilitates positive outcomes for patients.	Qualitative: FGDs and semi-structured interviews	Service users (n = 96)	‘Unmet social needs’, including individuals with anxiety and depression	Majority of participants described as low income	Social prescribing: link workers 11 community health centres; referred to activities by health provider directly or via link worker. Supported to attend prescribed with opportunity to become involved as ‘health champion’.	High
Carnes et al., 2015 [40] City and Hackney, London, England	To assess the social prescribing project with respect to effects on individuals, primary care team awareness of community resources, and costs.	Mixed methods (evaluation): interviews, FGDs, field visits, quantitative survey, electronic patient record data	Coordinators (n = 4), GPs (n = 2), commissioners (n = 2), service users (n = 5), 4 field visits to service providers	Social isolation, including individuals with anxiety and depression	Area of socioeconomic deprivation	Social prescribing: link workers 22 GP practices; patients referred to social prescribing coordinator, discussed action plans for achieving goals in up to 6 sessions.	Low
Chng et al., 2021 [41] Glasgow, Scotland	To explore the implementation process of the link worker approach to social prescribing in practice.	Qualitative: FGDs, email surveys, in-depth interviews	GPs, link workers, practice managers and community organisation workers (n = 31 FGDs, 19 surveys, 33 in-depth interviews, staff distribution unclear)	Social problems and multimorbidity, including individuals with anxiety and depression	Areas of socioeconomic deprivation	Social prescribing: link workers ‘Deep End Link Worker Programme’ in seven GP practices in deprived areas of Glasgow. Link workers embedding social prescribing approaches within practices, establishing referral pathways, and building networks with local community organisations.	High
Friedli et al., 2012 [42] Dundee, Scotland	To evaluate a pilot social prescribing scheme.	Mixed methods (evaluation): Pre- and post-intervention questionnaires, link worker notes and reflections, qualitative interviews, semi-structured interviews	Service users (n = 16), GPs (n = 2), link workers (n = 3)	Social problems and long-term conditions including mild to moderate anxiety and depression	Area of socioeconomic deprivation	Social prescribing: link workers One GP surgery (‘test site’); patients referred to link worker, had up to four consultations for assess needs and identify community-based information, support or activities.	Low
Galvin et al., 2000 [43] Southern county of England	To explore the impact of a Citizens Advice Bureaux service in primary care.	Mixed methods: quantitative questionnaire; qualitative interviews; FGDs	Service users (n = 10), CAB advisers (n = 2), GPs (n = 6)	‘Complex needs’ including depression	Service mainly accessed by people needing help with financial problems, accessing benefits or having employment issues.	Social prescribing Citizens Advice Bureau advisers located in GP practices to offer advice e.g., on finance, housing and employment.	High
Hanlon et al., 2021 [44] Glasgow, Scotland	To investigate if Self Determination Theory can be used to understand the change, or lack of change, resulting from patients’ involvement in the Links Worker Programme.	Qualitative: Semi-structured interviews (conducted as part of quasi experimental cluster RCT)	Service users (n = 12)	‘Complex needs’ including anxiety and depression	Areas of socioeconomic deprivation	Social prescribing: link workers 11 GP practices; one community link practitioner attached to each surgery. Patients referred to community link practitioners to signpost and support engagement with community organisations. Involved GP surgeries had a practice development fund and access to shared learning events for practice staff.	High
Hassan et al., 2020 [45] Northwest of England	To explore a community hub project as a social prescribing model and identify key elements that contribute toward enhancing its effectiveness.	Qualitative: FGDs	Service users (n = 18)	‘Mental health’	Areas of socioeconomic deprivation	Social prescribing “Life Rooms” programme: physical space for learning opportunities and social support from specialist staff, and peer tutors. Provides safe environment including cafe and computer facilities.	High
Kiely et al., 2021 [46] Ireland	To report the results of a pilot study, conducted in preparation for an RCT that aims to test the effectiveness of primary care-based link workers.	Qualitative: Structured interviews	Service users (n = 6), link worker (n = 1), GPs (n = 2)	‘Individuals with multimorbidity’; including those with anxiety and depression	Area of socioeconomic deprivation	Social prescribing: link workers One pilot GP practice; patients invited to meet with link workers at the practice link workers supported them over six week period to access recommended community resources.	Medium
Knowles et al., 2015 [47] Northwest of England	To report the results of a nested qualitative study within a trial which aimed to examine how a collaborative care model was implemented, and how providers experienced the integration of physical and mental healthcare.	Qualitative: semi-structured interviews (conducted as part of wider trial)	Service users (n = 31), practice nurses (n = 12), psychological wellbeing practitioners (n = 11), GPs (n = 7).	Depression and long-term conditions	Areas of socioeconomic deprivation	New method of clinical practice 17 GP practices; collaborative care integrating depression care within the management of long-term conditions. Eight sessions of low intensity psychological therapy delivered by psychological wellbeing practitioners who acted as case managers, including joint meetings with practice nurses and discussion of physical-mental comorbidities.	High
Makin and Gask, 2012 [48] Salford, England	“To explore the added value of participation in an Arts on Prescription programme to aid the process of recovery in people with common but chronic mental health problems that have already undergone a psychological ‘talking’-based therapy”.”	Qualitative: in-depth interviews	Service users (n = 15)	Mild to moderate anxiety and depression	Area of socioeconomic deprivation	Social prescribing ‘Arts on prescription’: referrals from health/social care providers, initial assessment by mental health worker, two arts sessions weekly for six months	High
Moffat et al., 2017 [49] Newcastle-upon-Tyne, England	To describe the experiences of patients with long-term conditions in a Link Worker social prescribing programme and identify the programme’s impact on health and wellbeing.	Qualitative: semi-structured interviews	Service users (n = 30)	Long-term conditions, including individuals with co-morbid anxiety and depression	Area of socioeconomic deprivation	Social prescribing: link workers ‘Ways to Wellness’ programme: referrals from primary care practitioner, link workers trained in behaviour change meet with them to identify goals and connect to community groups and resources.	High
Mulligan et al., 2020 [50] Ontario, Canada	To examine how a social prescribing programme was implemented, perceptions of the programme, and its impact on healthcare systems.	Mixed methods (evaluation): FGDs, aggregate-level data reporting	Service users, community champions, providers (numbers in FGDs unclear)	‘Unmet social needs’, including individuals with anxiety and depression	Majority of participants described as low income	Social prescribing: link workers This is an early report which provides a descriptive summary of main outcomes. It precedes the journal article Bhatti 2021 and reports on the same project (see above).	Low
Payne et al., 2020 [51] Sheffield, England	“To explore the ways by which social prescribing may be beneficial to individuals undertaking socially prescribed activity.”	Qualitative: semi-structured interviews	Service users (n = 17)	Range of psychosocial issues, including depression	Areas of socioeconomic deprivation	Social prescribing: link workers Social prescribing organisation: referrals from local healthcare professionals, initial consultation then signposted to community groups or within the organisation to advocacy, social cafes and health trainers.	High
Pescheny et al., 2018 [52] Luton, England	“To explore the experiences and views of service users, involved GPs, and navigators on factors influencing uptake and adherence to social prescribing”.	Qualitative: semi-structured interviews	Service users (n = 10), GPs (n = 3), navigators (n = 2)	Range of psychosocial issues, including anxiety and depression	Area of socioeconomic deprivation	Social prescribing HCP refers to navigator. Navigators can refer service users onwards to a maximum of 12 sessions e.g., meditation, physical activity, arts, job centre support, housing advice	High
Siantz et al., 2020 [53] Southern California, USA	To explore patient experience and provider perceptions within an initiative to deliver integrated care to people with complex needs as part of a system redesign.	Qualitative: FGDs, qualitative interviews	Service users (n = 54), providers (n = 32)	‘Complex needs’, including depression	Individuals using Medicaid services (health coverage for people on low income)	New model of care Behavioral Health Integration and Complex Care Initiative (BHICCI): a large-scale system redesign to deliver integrated care to people with complex needs. Delivery across multiple participating health care clinics. Participating clinics created multidisciplinary teams to oversee care management of patients with >1 chronic care condition and one mental health condition.	High
Sinclair, 2017 [54] Glasgow, Scotland	To explore the impact of an advice worker project in GP practices both on people delivering and accessing the project, and to better understand the supporting processes.	Mixed methods (evaluation): semi-structured interviews, written notes during meetings and conversations, referral and financial data	GPs (n = 3), advice worker (n = 1), practice managers (n = 2)	Self-reported long-term illness, mobility impairments and ‘mental health issues’	Areas of socioeconomic deprivation	Social prescribing Advice workers delivering service in two GP practices, offering appointments on issues such as housing, social security and financial management. If appropriate, they referred people onto additional forms of specialist community support, such as carers’ support, mental health and homelessness organisations.	Medium
Skivington et al., 2018 [55] Glasgow, Scotland	“To investigate issues relevant to implementing a social prescribing programme to improve inter-sectoral working to achieve public health goals.”	Qualitative: semi-structured interviews	Community link practitioners (n = 6), community organisation representatives (n = 30)	Community organisations designed for range of problems including ‘mental health’	Areas of socioeconomic deprivation	Social prescribing: link workers ‘Links Worker Programme’: primary care professionals refer to community links practitioners, who support patients to access community organisations. It is also designed to improve inter-sectoral working by strengthening links between GP practices and community organisations.	High
Webster et al., 2016 [56] North of England	“To examine the acceptability of a Brief Behavioural Activation intervention within a collaborative care framework.”	Qualitative: semi-structured interviews (nested qualitative evaluation within a service development pilot)	GPs (n = 5), practice nurse (n = 3), healthcare assistant (n = 1), mental health gateway worker (n = 1), service users (n = 4)	Depression and long-term conditions	Area of socioeconomic deprivation	New method of clinical practice A ‘collaborative care framework’ to address both depression and long-term conditions. Practice nurses acted as case managers and delivered Behavioural Activation (intervention for depression), with support from GPs and a mental health specialist.	High
White et al., 2017 [57] West Central Scotland	To “examine factors that may promote or compromise the implementation of social prescribing, as a collaboration between statutory and third sectors in Scotland”.	Qualitative: semi-structured interviews	Health visitors (n = 7), district nurses (n = 8), GPs (n = 3), community organisation representatives (n = 15)	‘Mental health’	Areas of socioeconomic deprivation	Social prescribing Three community organisations: a healthy lifestyle project, a carers’ centre and a project to address mental health issues including substance misuse. Service users were referred to these programmes by health visitors, district nurses and GPs.	High
Wildman et al., 2019 [58] Newcastle-upon-Tyne, England	“To explore experiences of social prescribing among people with long-term conditions one to two years after their initial engagement with a social prescribing service”.	Qualitative: semi-structured interviews Follow up study of participants from Moffatt 2017.	Service users (n = 24)	‘Mental health and social isolation issues’, including anxiety and depression	Area of socioeconomic deprivation	Social prescribing: link workers ‘Ways to Wellness’ programme: referrals from primary care practitioner, link workers trained in behaviour change meet with them to identify goals and connect to community groups and resources.	High
Wildman et al., 2019 [59] Newcastle-upon-Tyne, England	To explore how link workers understand their role in social prescribing, as well as its challenges and threats to client engagement.	Qualitative: FGDs, semi-structured interviews	Link workers (n = 26)	Long-term conditions, including individuals with co-morbid anxiety and depression	Area of socioeconomic deprivation	Social prescribing: link workers ‘Ways to Wellness’ programme: referrals from primary care practitioner, link workers trained in behaviour change meet with them to identify goals and connect to community groups and resources.	High

## Data Availability

Not applicable.

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
