# Peer review of "Experiences of Non-Pharmaceutical Primary Care Interventions for Common Mental Health Disorders in Socioeconomically Disadvantaged Groups: A Systematic Review of Qualitative Studies"

_ijerph, 2023, doi:10.3390/ijerph20075237_

Round 1
Reviewer 1 Report
It's an interesting piece of work. The language expression and methodological process are smooth and normative. In addition, considering this article is a less common systematic review of qualitative studies in this field and was completed in conjunction with another published review of quantitative studies, it deserves to be considered for publication so that readers can gain a deeper understanding of this area, although the findings were slightly simpler (I would suggest Fig.2 put more effort into the presentation of the mechanism).
Reviewer 2 Report
This is a nicely done study, very clearly presented. Some of the findings might be further emphasized. For example, uncertainty about funding might be particularly harmful to these patents. Advocates might pressure lawmakers to provide sustained funding for these programs. The authors might offer some political background for the implications of their study.
Specific comments:
Line 18--impact on
l. 85--engagement of, not with
l. 98--impact on
l. 111--was, not were
Under heading Sense of control..., line 2--engaging with
Under heading Confidence..., line 1--self-confidence
Under heading Understanding of health..., line 4th from bottom: Impacted upon
Under heading Functioning..., paragraph 2, line 4th from bottom: engaging with
Under heading Positive relationships, para. 3, line 1---engagement with
Under heading3.2.3...,para 1, line 2--were, not was; line 3--engaging with; para 4, line 1--engagement with
Under heading Feeling failed..., para 1, last line--omit in
Under Discussion
para 1 line 3--be, not were
para 3, 4th last line--centring is an uncommon word in American English
para 4, line 5--engagement with
Heading 4.1...,para 1, line 5--engagement with; para 3, line 5th from bottom--engage with
Heading Implications--para 1, line 3--engage with; para 4, bullet point 1--again, centred will baffle most Americans
Heading Conclusions, para 1, line 5--engage with
Instead of repeatedly using engage with, which is grammatically incorrect and a cliche, how about "use" or "utilize"?
Reviewer 3 Report
Dear authors,
Thank you for your manuscript on this important topic. The manuscript is overall coherent and meets the requirements of the journal. With regard to the methodology, it is particularly noteworthy that the authors have both published a study protocol and registered the review. Furthermore, the review was prepared according to the PRISMA guideline. For the qualitative studies included, the authors could have additionally included the work of Tong et al. (2021) "Enhancing transparency in reporting the synthesis of qualitative research: ENTREQ". I particularly like the detailed table with the overview of the included studies and the assessment with the CASP tool. With regard to strengths and limitations, it could have been helpful to mention that the interventions described are usually complex interventions.
